# Nature’s Role in Supporting Health during the COVID-19 Pandemic: A Geospatial and Socioecological Study

**DOI:** 10.3390/ijerph18052227

**Published:** 2021-02-24

**Authors:** Jake M. Robinson, Paul Brindley, Ross Cameron, Danielle MacCarthy, Anna Jorgensen

**Affiliations:** 1Department of Landscape Architecture, The University of Sheffield, Sheffield S10 2TN, UK; p.brindley@sheffield.ac.uk (P.B.); r.w.cameron@sheffield.ac.uk (R.C.); a.jorgensen@sheffield.ac.uk (A.J.); 2inVIVO Planetary Health of the Worldwide Universities Network, West New York, NJ 10704, USA; 3The Healthy Urban Microbiome Initiative (HUMI), Adelaide, SA 5005, Australia; 4School of Natural and Built Environment, Queen’s University Belfast, Belfast BT9 5AG, UK; Dmaccarthy01@qub.ac.uk

**Keywords:** COVID-19, coronavirus, green space, planetary health, nature connectedness, public health, nature-based interventions

## Abstract

The COVID-19 pandemic has brought about unprecedented changes to human lifestyles across the world. The virus and associated social restriction measures have been linked to an increase in mental health conditions. A considerable body of evidence shows that spending time in and engaging with nature can improve human health and wellbeing. Our study explores nature’s role in supporting health during the COVID-19 pandemic. We created web-based questionnaires with validated health instruments and conducted spatial analyses in a geographic information system (GIS). We collected data (*n* = 1184) on people’s patterns of nature exposure, associated health and wellbeing responses, and potential socioecological drivers such as relative deprivation, access to greenspaces, and land-cover greenness. The majority of responses came from England, UK (*n* = 993). We applied a range of statistical analyses including bootstrap-resampled correlations and binomial regression models, adjusting for several potential confounding factors. We found that respondents significantly changed their patterns of visiting nature as a result of the COVID-19 pandemic. People spent more time in nature and visited nature more often during the pandemic. People generally visited nature for a health and wellbeing benefit and felt that nature helped them cope during the pandemic. Greater land-cover greenness within a 250 m radius around a respondent’s postcode was important in predicting higher levels of mental wellbeing. There were significantly more food-growing allotments within 100 and 250 m around respondents with high mental wellbeing scores. The need for a mutually-advantageous relationship between humans and the wider biotic community has never been more important. We must conserve, restore and design nature-centric environments to maintain resilient societies and promote planetary health.

## 1. Introduction

The 2020 COVID-19 pandemic has brought about unprecedented changes to human lifestyles across the world. This includes considerable disruptions to urban mobility patterns and social interactions [1,2]. In many countries, governments have imposed ‘lockdowns’ and other ‘social distancing’ restrictions to reduce the transmission and spread of the SARS-CoV-2 virus and prevent the collapse of health services [3,4]. However, evidence suggests that these social restrictions are associated with higher rates of negative mental health outcomes such as depression, insomnia [5], suicidal ideation [6], and anxiety [7].

Although not a panacea, the importance of spending time in and engaging with natural environments such as parks and woodlands for physical and mental health is well documented. For example, shinrin-yoku (森林浴) or ‘forest bathing’ has been shown to reduce blood pressure and anxiety [8]. Urban nature supports mental health and wellbeing [9], and access to a garden is associated with higher levels of wellbeing [10]. Furthermore, green spaces can harbour diverse microorganisms [11] that transfer to humans after a short period of time spent in these environments [12]. Importantly, exposure to a diverse range of microorganisms from the environment can regulate the human immune system [13].

Recent studies have demonstrated that patterns of visiting natural spaces such as urban parks and woodlands have changed as a result of COVID-19 [2,14]. Other studies have called for keeping parks and green spaces accessible during the COVID-19 pandemic due to their health benefits [15,16]. A recent study showed that participation in some nature-based activities increased (e.g., foraging, gardening, hiking, jogging, and watching wildlife), while others decreased (e.g., camping) [17]. Although these studies have commented on the potential health and wellbeing benefits of engaging with nature during the pandemic, to our knowledge, few have specifically explored the multifaceted benefits on mental health and wellbeing using validated research instruments. Furthermore, no studies have explicitly investigated how socioecological factors such as deprivation (e.g., based on economic factors, crime risk, education and living environment), access to green spaces, and vegetation cover may influence health and wellbeing outcomes.

In this mixed-methods study, we provide a comprehensive analysis of the effects of natural spaces (e.g., parks, woodlands, lakes) on self-reported health and wellbeing. We also investigate aspects of changes to patterns of nature exposure, and potential socioecological drivers of wellbeing outcomes by collecting data on pre and during COVID-19 perceptions using a single sampling time point. We use online pilot-tested questionnaires with validated wellbeing instruments including the 14-item Warwick–Edinburgh Mental Wellbeing Scale (WEMWBS) and the 10-item Perceived Stress Scale (PSS). To assess nature connectedness (one’s affective, cognitive, and experiential connection with the natural world) [18], we used the 6-item Nature Relatedness scale (NR-6). We also used a geographic information system (GIS) to study how socioecological factors including deprivation (as defined above), presence/abundance of green spaces, and relative greenness may affect wellbeing outcomes.

The primary objectives of this study were to: (a) assess whether people’s patterns of exposure to nature changed as a result of the COVID-19 pandemic (and to characterise these changes); (b) assess whether nature provided a health and wellbeing benefit during the pandemic (and to characterise these benefits); and (c) investigate whether potential health outcomes were significantly affected by socioecological factors such as deprivation, the presence and abundance of green spaces, and relative greenness.

Gaining a better understanding of how socioecological factors affect human health and wellbeing during the COVID-19 pandemic will help to inform environmental management and public health policy. This study also provides important information on how populations respond to emerging infectious disease pandemics and how we can optimise the mitigation of the associated negative impacts. This knowledge will be increasingly valuable as the number and diversity of human infectious diseases outbreaks have increased since 1980 [19]. Moreover, pandemics are expected to increase in frequency in the future [20]. Indeed, the projected increase in global urbanisation has the potential to augment hazardous interfaces for zoonotic pathogen exposure [21].

Natural environments should be conserved and restored on a global level, but also maintained and promoted at the urban/community level to support health and wellbeing, not just in the face of emerging pandemics, but to maintain healthy societies overall [22].

## 2. Methods

### 2.1. Study Design and Participants

#### 2.1.1. Digital Questionnaire and Validated Wellbeing Instruments

We created a web-based research questionnaire using the Smart Survey online platform [23]. The questionnaire included 52 multi-format questions (Appendix A) aimed at measuring different aspects of mental wellbeing and nature connectedness. To measure wellbeing, we used the 14-item Warwick–Edinburgh Mental Wellbeing Scale (WEMWBS) [24,25,26]. Between April and July 2020, we asked participants to answer questions regarding their wellbeing in recent weeks, as well as in the weeks prior to the COVID-19 pandemic. The WEMWBS includes 14 items, on a 1–5 Likert scale relating to perceived state of mental wellbeing (emotional and cognitive). The continuous scale was scored by summing the responses to each item answered, ranging from 14 (lowest possible wellbeing score) to 70 (highest possible wellbeing score). We measured perceived stress using the 10-item Perceived Stress Scale (PSS) [27,28]. The PSS measures the degree to which one feels stressed by evaluating coping recourses and feelings of control. We asked participants to answer questions regarding perceived stress in recent weeks, as well as in the weeks prior to the COVID-19 pandemic. The PSS includes 10 items, on a 1–5 Likert scale. The PSS scores range from 0 (lowest possible stress score) to 40 (highest possible stress score), and higher scores indicate higher levels of perceived stress. We also measured nature connectedness using the Nature Relatedness Scale (NR-6) [29,30]. The NR-6 includes 6 items, on a 1–5 Likert scale, and presents questions such as “I feel very connected to all living things and the earth” and “my relationship to nature is an important part of who I am”. Items were averaged, and higher scores indicated stronger subjective connectedness to nature. All of the validated instruments used in this study have been used in previous green space epidemiology studies [31,32,33]. We also asked several pilot-tested questions regarding nature exposure such as duration and frequency of visits, environment type, and reasons for visits (Appendix A).

The questionnaire received ethics approval by the University of Sheffield’s Department of Landscape Architecture’s internal review committee. The questionnaire also requested key demographic information including age, gender, location (postcode), highest level of education, and occupation. The questionnaire was distributed across the world (between April and July 2020) via a secure weblink with a detailed participant information sheet, consent form and the questionnaire. However, most of the recipients were in the UK due to global distribution limitations. We used a range of non-random sampling approaches to reach potential participants including emailing volunteer groups (mostly in the UK due to search constraints/time limitations), posting on social media such a Twitter and LinkedIn (which reached participants from the UK, USA, Canada, Australia, India, China, Brazil, Argentina, Portugal, Germany, Nepal, New Zealand, and South Africa), and undertaking a web scrape of publicly available community group directories, and emailing the group leaders (again, mostly in the UK) (Appendix A). People under the age of 18 years were not included in this study (the only exclusion criterion except for the geospatial analysis section, where only England-based responses were analysed due to sample size and appropriate GIS datasets).

#### 2.1.2. Geospatial Analysis

We cleaned the spreadsheet containing the responses and geolocations, imported it into QGIS 3.4 (QGIS Development Team, 2002) as a comma separated value (.csv) vector layer, and converted it to an ESRI point shapefile. Using vector geoprocessing tools, buffer radii of 50, 100, 250, and 500 m were generated around each point (respondent’s postcode) to facilitate spatial analyses (Figure 1). Similar buffers have been used in previous geospatial and socioecological studies [34,35]. To explore green space presence and abundance, we imported the OS Open Greenspace dataset (publicly accessible urban green spaces in the UK) into QGIS as a polygon vector layer. These datasets have been used in several urban socio-ecological studies [36,37]. Figure 1 highlights the concept of buffer and green space analysis used in this study. We also imported UK forest shapefiles (>5 ha) from the National Forest Inventory (Forestry Commission, 2020) using the same methods.

To acquire a measure of mean greenness for each buffer radius, we used the Copernicus Sentinal-2 satellite imagery (10 m resolution), downloaded from the EDINA Digimap Ordnance Survey Service [38]. We isolated spectral bands 4 (Red) and 8 (Near Infrared) and applied the following equation for the Normalised Difference Vegetation Index (NDVI):$$\frac{\mathrm{Near}\;\mathrm{Infrared}\;\mathrm{Light}-\mathrm{Red}\;\mathrm{Light}}{\mathrm{Near}\;\mathrm{Infrared}\;\mathrm{Light}+\mathrm{Red}\;\mathrm{Light}}$$

This equation provides a score of estimated land-cover greenness, whereby 0 represents a very low level of greenness and 1 represents a very high level of greenness. The greenness score can be used as a proxy for vegetation biomass and cover [39,40]. We used the algebraic expression calculator in QGIS to process the raster files (the two Sentinel-2 spectral band layers: red and near infrared). We then calculated the mean NDVI values for all buffer zones using the zonal statistics raster analysis tool. The attribute table was then exported as a .csv file. This enabled downstream analysis in R (version 4.0.2; R Core Development Team, Vienna, Austria).

#### 2.1.3. Deprivation

To explore relative deprivation, we calculated quintile scores from the 2019 index of multiple deprivation (IMD) dataset. The IMD has been used in a range of epidemiology and urban ecology studies [41,42]. In England, the IMD provides an output of relative deprivation based on multivariate analysis of demographic data (e.g., economics, crime risk, education and living environment) acquired for Lower Super Output Areas (LSOAs) [43]. LSOAs are a geographic hierarchy designed for the reporting of small area statistics. The LSOA boundaries represent an average population of approximately 1500 and have been used widely in socioecological studies [44,45].

### 2.2. Statistical Analysis

To assess proportional differences between pre/during COVID-19 patterns of exposure to nature we used 2-sample tests for equality of proportions with continuity corrections in R. We used one sample t-tests to compare differences in mean frequency of visits and duration of time spent in nature before and during the pandemic. We applied the one-way analysis of variance (ANOVA) test to determine whether socioeconomic status (based on IMD) affected the mean frequency of visits and duration of time spent in nature before and during the pandemic. A binomial regression generalised linear model (GLM) was used to explore responses to environmental preferences, and point estimates were used to indicate which environments were associated with the greatest odds for visits.

To analyse self-reported wellbeing and perceived stress, the WEMWBS and PSS scores were recoded into binary variables by division into high and lower scores. For WEMWBS, we used scores of 60+ as an indication of high wellbeing [46]. For the PSS, we used scores of 16+ as an indication of high stress [47]. We built logistic regression models to investigate relationships between wellbeing, perceived stress and different ecological variables including green space presence and abundance, forest presence and abundance, and vegetation cover/greenness (via NDVI). An odds ratio (OR) of 1 or above means the predictor variable increases the odds of scoring a high level of wellbeing. An OR < 1 means the predictor variable decreases the odds of scoring a high level of wellbeing (and the same for perceived stress). We applied model adjustments for gender, age, socioeconomic status, level of education, work/living situation, and nature connectedness. We repeated these models for each buffer area (50, 100, 250, and 500 m).

We also examined associations between nature connectedness and duration of nature visits, frequency of visits to nature per week, and self-reported wellbeing via the WEMWBS. We applied Pearson’s product–moment correlation test. Using the psych [48] and boot [49] packages in R, we applied bootstrap resampling to assign a measure of accuracy to sample estimates for correlations with a minimum of 1000 iterations.

## 3. Results

A total of *n* = 1184 respondents completed our research questionnaire. We acquired a broad distribution of responses, predominantly (*n* = 993 or 96% of georeferenced responses) from across England, UK (Figure 2B). We also acquired complete datasets for green spaces, IMD, and forests (>0.5 ha) for England to conduct the geospatial analysis (Figure 2A,C,D). Therefore, in the geospatial analysis, only responses from England were included. There was a skew towards respondents who identified as being female (*n* = 851 or 72%) compared to male (*n* = 331 or 28%), trans woman (*n* = 1 or 0.1%), and non-binary (*n* = 1 or 0.1%), and towards respondents with a higher level of education (*n* = 847 or 72% with ≥ undergraduate degree). Taking the median age category, the distribution either side was similar (*n* = 624 or 53% were 55 years old or over; and *n* = 560 or 47% were 54 years old or younger). This is not quite representative of the UK age structure (~31% are over 55 years old) [50].

### 3.1. Changing Patterns of Exposure to Nature during the COVID-19 Pandemic

Our results show that a significantly greater proportion (88%) of participants reported that they spent more time in natural environments as a result of the COVID-19 pandemic, confirmed by a 2-sample test for equality of proportions with continuity correction (*X*_2_ = 1525, df = 1, *p* = <0.01). This was in contrast to those who reported spending less time in nature (7%) and those whose reported patterns of exposure did not change (5%). Table 1 shows a breakdown of the most popular responses. The most popular environments (based on a duration increase) were private gardens (47.7%), followed by woodlands (13.7%), and urban parks (10.9%). Over 80% of all participants reported that they were likely to spend more time in nature once the COVID-19 pandemic is over, which is also a significant proportional difference (*X*_2_ = 853, df = 1, *p* = <0.01).

The average reported duration that participants spent in natural environments increased during the COVID-19 pandemic (x = 106 min) compared to before the pandemic (x = 66 min), and was statistically significant (t = −15.491, df = 2310.8, *p* = <0.01) (Figure 3A). The average reported frequency of visits to natural environments per week also increased during the COVID-19 pandemic (x = 5 visits) compared to before the pandemic (x = 4 visits), and was also statistically significant (t = −4.8263, df = 2336, *p* = <0.01) (Figure 3B).

Our results show that IMD did not significantly affect the reported duration spent in nature before or during the pandemic (ANOVA, df = 4, F = 0.74, *p* = 0.6; and df = 4, F = 0.55, *p* = 0.7, respectively). Furthermore, IMD did not significantly affect the reported frequency of visits to nature per week before or during the pandemic (ANOVA, df = 4, F = 1.5, *p* = 0.2; and df = 4, F = 1.1, *p* = 0.3, respectively). Gender did not significantly affect the reported duration or frequency (ANOVA, df = 2, F = 0.5, *p* = 0.5). We confirmed these non-significant relationships for each IMD quintile with a Tukey multiple comparison of means test.

The ANOVA test results showed that the reported duration of nature visits before the pandemic was significantly different depending on age (ANOVA, df = 7, F = 2.3, *p* = 0.02). However, the Tukey multiple comparison of means test showed that differences were only significant between 75 and 84 years old and both 45–54 years old (x difference = +26 mins, *p* = 0.02) and 55–64 years old (x difference = +23 mins, *p* = 0.04). In other words, the 75–84 years old reportedly spent more time per visit to nature than 45-64 years old before the pandemic. However, there were no significant differences in reported duration between any age group during the pandemic (ANOVA, df = 7, F = 1.375, *p* = 0.2). There were also no significant differences in frequency of visits per week between any age group before the pandemic (ANOVA, df = 7, F = 1.2, *p* = 0.3) or during the pandemic (ANOVA, df = 7, F = 0.4, *p* = 0.9).

There was a statistically significant difference in responses to the question “Are there any outdoor environments that you would be concerned to visit as a result of COVID-19?” (GLM_Binomial_, *X*_2_ = 743, df = 6, *p* = <0.01). Point estimates indicate that beaches and urban parks are associated with the greatest odds for (perceived) concern due to COVID-19 (Figure 4). This implies that concern for contracting SARS-CoV-2 virus may influence people’s decision to spend time in certain environments.

We also show that 34% of participants reportedly visited natural environments that they would not usually visit as a result of the COVID-19 pandemic. There was a statistically significant difference in responses (GLM_Binomial_, *X*_2_ = 22, df = 11, *p* = 0.02), and point estimates indicate that woodlands (56% of responses) are associated with the greatest odds for novel visits (Figure 5).

### 3.2. Nature’s Influence on Health and Wellbeing during the COVID-19 Pandemic

Overall, respondents’ self-reported mental wellbeing reduced significantly (t = 19.1, df = 2349, *p* = <0.01) during the pandemic compared to before the pandemic. Interestingly, mean perceived stress scores were slightly lower during the pandemic compared to before the pandemic (t = 1.9, df = 2305, *p* = 0.05). However, mean perceived stress scores before and during the pandemic were both in the highest PSS scoring range (Table 2). Of the respondents whose duration in nature increased during the pandemic (*n* = 911), a significantly greater proportion showed decreased perceived stress (*X*^2^ = 8, df = 1, *p* = <0.01). Moreover, of the respondents whose perceived frequency of visits to nature increased during the pandemic (*n* = 632), a significantly greater proportion showed decreased perceived stress (*X*^2^ = 5.5, df = 1, *p* = 0.01). Furthermore, when comparing people’s work/living situation, there was only a significant difference in perceived stress levels before and during the pandemic for those who were “furloughed or unemployed as a result of COVID-19”. Their perceived stress levels were significantly lower during the pandemic (t = 2.4, df = 350, *p* = 0.01).

Eighty-four percent (*n* = 1004) of respondents agreed that spending time in nature helped them cope with the COVID-19 pandemic, and 56% (*n* = 569) of these ‘strongly agreed’. When comparing the responses for male and female we found a significant difference in the strength of respondents’ agreement (W = 17,060, *p* = <0.01). The median female score was 7 (strongly agree), while the median male score was 6 (agree). We also found that the strength of respondents’ agreement was significantly different depending on their living situation (H = 14.357, df = 6, *p* = 0.02). For example, the median score for participants “at home and not working due to being furloughed or unemployed as a result of COVID-19” (*n* = 211) was 7 (strongly agree), and the median score for those working (either at home or still at their workplace) (*n* = 564) was 6 (agree) (Figure 6).

There were also differences in the perceived ways in which nature helped respondents cope with COVID-19 (GLM_Binomial_, *X*_2_ = 1138, df = 6, *p* = <0.01) (Table 3A). The most popular response was that nature provided a place to exercise (x = 0.7), followed by helping to reduce stress (x = 0.6) and providing a calm space to think (x = 0.58).

Ninety-seven percent of participants (*n* = 397) who reportedly visited novel (to the respondent) natural environments as a result of COVID-19 did so for a health and wellbeing benefit. There were significant differences in terms of popularity of responses (GLM_Binomial_, *X*_2_ = 836, df = 8, *p* = <0.01). Physical exercise (x = 0.3) and fresh air (x = 0.3) followed by relaxation (x = 0.23) were the top three most popular perceived nature-mediated benefits (Table 3B).

There was no significant association between level of nature connectedness and self-reported mental wellbeing before the pandemic, as shown by a bootstrap-resampled Pearson’s correlation (*r* = 0.05, df = 1179, *ß* = 0.05 (−0.01–0.11), *p* = 0.13). However, level of nature connectedness did show a weak but significant association with self-reported mental wellbeing during the pandemic (*r* = 0.07, df = 1179, *ß* = 0.07 (0.02–0.13), *p* = 0.01). When we compared the scores for females and males, we found that the association between nature connectedness and self-reported mental wellbeing before the pandemic was not significant for females (*r* = 0.01, df = 849, *ß* = 0.01 (−0.05–0.08), *p* = <0.74) and the association during the pandemic was also not significant (*r* = 0.04, df = 849, *ß* = 0.04 (−0.02–0.12), *p* = <0.16). However, the association between nature connectedness and mental wellbeing before the pandemic was significant and stronger for males (*r* = 0.12, df = 328, *ß* = 0.12 (0.01–0.24), *p* = 0.02), and the association during the pandemic was also significant (*r* = 0.13, df = 328, *ß* = 0.13 (0.02–0.24), *p* = 0.02).

The correlation results also show there was a weak but significant positive association between frequency of visits to natural environments and level of nature connectedness (*r* = 0.12, df = 991, *ß* = 0.12 (0.06–0.19), *p* = <0.01). We also show a significant positive association between duration of visits to natural environments and level of nature connectedness (*r* = 0.17, df = 991, *ß* = 0.17 (0.11–0.23), *p* = <0.01). However, when comparing scores for female and males, the association between nature connectedness and duration in nature for females was not significant (*r* = 0.00, df = 708, *ß* = 0.00 (−0.07–0.07), *p* = 0.95). The association between nature connectedness and frequency of nature visits was also not significant (*r* = 0.00, df = 707, *ß* = 0.00 (−0.06–0.08), *p* = 0.83). The association between nature connectedness and duration in nature for males was not significant (*r* = 0.03, df = 280, *ß* = 0.03 (−0.08–0.16), *p* = 0.53). The association between nature connectedness and frequency of nature visits was also not significant for males (*r* = 0.04, df = 280, *ß* = 0.04 (−0.08–0.14), *p* = 0.53).

### 3.3. The Relationship between Health Outcomes and Spatial/Socioecological Factors

Our results show that 94% (*n* = 1118) of the survey responses came from the UK. Of these respondents, 92% (*n* = 1031) provided georeferenced identifiers (in the form of postal codes). Ninety-six percent (*n* = 993) of these respondents were based in England. Therefore, *n* = 993 responses were included in the logistic regression models built to investigate potential relationships between green space, NDVI, mental wellbeing and perceived stress. This enabled a standardised analysis of socioeconomic status via the IMD (unique to England).

The results from our unadjusted logistic regression models show that there was a significant positive effect of NDVI (greenness) on self-reported mental wellbeing in all of the spatial radii around a respondent’s home location (50, 100, 250, and 500 m). For the 250 m buffer, the significant positive effect of NDVI on self-reported mental wellbeing remained significant and with a relatively high odds ratio (>8) when adjusting for all of the covariates (OR: 8.04 (1.44, 45.01), *p* = 0.01).

However, in the 50, 100 and 500 m buffer radii (around a respondent’s home location), the significant effect remained only when adjusting for gender (OR: 4.92 (1, 24.13), *p* = 0.04; OR: 5.26 (1.03, 26.90), *p* = 0.04; OR: 5.2 (0.95, 29.3), *p* = 0.05, respectively) and not when adjusting for age (apart from the 65–74 year age range), socioeconomic status (IMD), nature connectedness, work/living situation and level of education (Table 4). The positive effect of NDVI on self-reported wellbeing was significant for the 65–74 year age range for both the 100 m buffer (OR: 4.49 (1.05, 19.22), *p* = 0.04) and the 500 m buffer (OR: 4.66 (1.09, 19.95), *p* = 0.03).

Our results also show no significant associations between green space (or forests—Appendix A) presence and abundance and self-reported mental wellbeing for any of the spatial buffers (Table 4).

In terms of perceived stress, there was a significant effect of NDVI on reducing stress in the 100 m (OR: 0.38 (0.15, 0.94), *p* = 0.03) and 250 m buffer zones (OR: 0.37 (0.14, 0.96), *p* = 0.04) with the unadjusted models (Table 5). In adjusted models, however, these significant levels tended to be lost; there being no other significant associations for NDVI, and green space presence on stress.

However, we further explored green space typology and found that within the 100 and 250 m buffer radii around a respondent’s postcode, the mean number of food-growing allotments was higher for those who had higher mental wellbeing scores (x = 0.07 and 0.31, respectively) compared to lower (x = 0.03 and 0.21, respectively). This was confirmed as a significantly greater proportion of allotments within 100 and 250 m of respondents with high mental wellbeing scores compared to low (*X*^2^ = 4.3 and 10.8, df = 1, *p* = 0.03 and <0.01, respectively). See Appendix A for a full breakdown of green space typologies.

## 4. Discussion

Our study shows that respondents perceived a significant change in their patterns of visiting nature as a result of the COVID-19 pandemic. People reportedly spent significantly more time in nature and visited nature more often during the pandemic. People generally perceived that they visited nature for a health and wellbeing benefit and the majority of respondents felt that nature helped them cope during the pandemic. Greater land-cover greenness within a 250 m radius around a respondent’s postcode was important in predicting higher levels of mental wellbeing. There were also significantly more food-growing allotments around respondents with higher mental wellbeing scores. This study provides an important contribution towards understanding how populations respond to infectious disease pandemics. It also further highlights the importance of conserving, restoring and designing nature-centric environments for human health and wellbeing.

As a result of the COVID-19 pandemic, over 90% of respondents reportedly increased the amount of time they spent in natural environments such as woodlands, parks, and gardens. Forty-eight percent of respondents reportedly spent more time in their private gardens. Fourteen percent of respondents reportedly spent more time in woodlands, and 11% spent more time in urban parks. People responded differently to the question “Are there any outdoor environments that you would be concerned to visit as a result of COVID-19?”. Beaches and urban parks were the environments that caused most concern with respect to visitations during the COVID-19 pandemic. This implies that concern for contracting SARS-CoV-2 virus influenced people’s decision to spend time in certain environments. Perhaps this is intuitive as beaches and urban parks traditionally attract crowds of people for recreational and social activities [51,52]. Moreover, there was considerable media coverage in the UK about overcrowding parks and beaches at the time, thus conceivably increasing the perceived risk of viral transmission. This information could be valuable to landscape managers and the public health sector. For example, understanding where additional anthropogenic pressures on the landscape (and upon sensitive ecological receptors) are likely to occur in response to pandemics could help with the formulation of appropriate mitigation measures. From an epidemiological perspective, comprehending patterns of behavioural change is also important for tracking and understanding disease dynamics [53,54].

Thirty-four percent of respondents also reportedly visited environments that they would not usually visit as a result of COVID-19. Our results indicate that woodlands were the most popular novel environment with 56% of these respondents visiting woodlands when they would usually not visit them. This further highlights the value of conserving and restoring woodlands and provides novel insights into human-environment interactions in the face of infectious disease pandemics.

Overall, respondents’ self-reported mental wellbeing reduced significantly during the pandemic. This corroborates other studies highlighting increases in anxiety [7], depression and insomnia [5] as a result of COVID-19. Interestingly, the slightly lower stress levels during the pandemic do not corroborate previous work [55]. We found that respondents who increased their duration and frequency of visits to nature, a greater proportion had lower perceived stress levels. This suggests that nature may provide a role in perceived stress relief and warrants further research. We also explored whether work/living situation affected the overall reduction in perceived stress and found an intriguing result. Only respondents who were furloughed or unemployed as a result of COVID-19 showed significantly lower stress levels during compared to before the pandemic (although both were still in the highest stress range). This could be due to a reduction in work-related stress, particularly for those who were furloughed and still receiving government-assisted payments. However, to fully understand these psychosocial dynamics, further research is warranted.

The majority of respondents agreed that spending time in nature helped them cope with the COVID-19 pandemic. This again highlights the immense value of conserving and restoring natural environments for human health and wellbeing. Perhaps in terms of our psychological resilience and ability to withstand disease pandemics, this has never been more salient. Indeed, the number and diversity of human infectious diseases outbreaks has increased significantly in the last 40 years [18]. Furthermore, as urbanisation continues to augment hazardous interfaces for zoonotic pathogen exposure [20], pandemics are expected to increase in frequency in the future [19]. Interestingly, when we compared the responses of females and males, we found a significant difference in the strength of respondents’ agreement to the statement “nature has helped me cope with the COVID-19 pandemic”. Indeed, the median female score was in the ‘strongly agree’ range, whilst the median male score was in the ‘agree’ range. This is corroborated by Roe et al. (2013), who showed that female stress levels (measured via cortisol sampling) were significantly higher in areas of low green space compared to males’, suggesting that there may be a gender difference in perceiving nature as a coping (or stress ameliorating) mechanism [56]. However, our results could also be the facet of the demographic limitations of this study, i.e., increasing the number of male respondents could potentially increase the median scores for males. This warrants a deeper investigation.

Ninety-seven percent of participants who visited novel natural environments—that is, novel to the respondent—as a result of COVID-19, reportedly did so for a health and wellbeing benefit. This suggests that people were actively seeking out new environments as a therapeutic response to COVID-19, and highlights the human appreciation for nature-centric features. The majority of respondents perceived natural environments as being important places for exercise, stress reduction and anxiety reduction. This corroborates broader wellbeing results from previous green space and epidemiological studies undertaken prior to the COVID-19 outbreak and during ‘non-pandemic’ times [9,10,57] and underscores the multifaceted benefits of engaging with nature. An interesting line of enquiry could be to compare the health outcomes and behavioural responses observed during the COVID-19 pandemic to non-pandemic times. Moreover, it would be interesting to see if COVID-19-mediated behavioural changes remain once the pandemic is over. Could we see positive nature-based outcomes arising from this unprecedented global experience—and could this contribute to a revived appreciation for nature and its associated health benefits?

Nature connectedness (one’s affective, cognitive, and experiential connection with the natural world) [17,58], which has previously been shown to associate with enhanced mental wellbeing [59,60], only associated with higher wellbeing before and during the pandemic for male participants. Further research is warranted to elucidate the reasons (and generalisability) for this gender difference and to ascertain the directionality of the association. Interestingly, our results show there was a significant positive association between frequency of visits and duration of visits to natural environments and level of nature connectedness. This supports the idea that spending time in and engaging with nature can increase one’s nature connectedness [61,62]. However, when analysing the results for females and males separately, the results were not statistically significant. This could be due to the p-value being a function of sample size as well as variance, and thus the reduction in sample size when stratifying the analysis may have affected the significance. Therefore, increasing the sample size would likely provide a richer and more accurate picture of the relationship between nature connectedness and duration/frequency of visits to nature.

Our results show that within the 250 m spatial buffer (around each respondent’s postcode), there was a significant positive effect of land-cover greenness on self-reported mental wellbeing during the COVID-19 pandemic. The relatively high odds ratio (>8) implies that a higher level of greenness (measured via the NDVI) significantly increases the odds of scoring a high level of wellbeing. This suggests that neighbourhood-scale greenery may be an important factor in the mental wellbeing of members of the community, which corroborates other studies [63,64]. There was no association in the 50, 100, or 500 m buffers, suggesting that very proximal land-cover greenness (e.g., in private gardens) and landscape greenness beyond the neighbourhood scale are potentially less important in moderating wellbeing. One possible explanation for this is that people may receive additional benefits from engaging with neighbourhood green spaces (such as safe, accessible, biodiverse parks with facilities) compared to gardens, street or isolated patches of vegetation with potentially smaller stature. Whereas 500 m green spaces may go beyond the neighbourhood scale and may be perceived by some as being less accessible. These results provide additional support for calls to augment neighbourhood vegetation cover and safe, accessible green spaces, and highlight the multidimensional benefits associated with urban greening. Future research should aim to develop innovative strategies that help to optimise the design, conservation and restoration of natural environments in urban areas where space is limited as a result of uncontrolled densification. For example, this could include enhancing street vegetation, creating small ‘pocket gardens’, changing vacant lots into mobile allotments, and incorporating bioreceptive materials into buildings.

When analysing publicly accessible green space as a single typology, there were no associations between these and mental wellbeing or perceived stress. These results could be affected by only having analysed the presence and abundance of green spaces and not fully considering their typology and quality (e.g., biodiversity, recreational potential, facilities, safety). For example, some of the OS green spaces include church yards (which many people may not visit), golf courses and bowling greens (often exclusive to members only). We did find that with deeper analysis, there were significantly more food-growing allotments within 100 and 250 m of respondents with higher mental wellbeing scores compared to lower. This again strengthens the calls for more quality and community-focused neighbourhood green spaces and urban gardens. As discussed, many people may have avoided parks due to overcrowding and the associated risks of contracting SARS-CoV-2. However, allotments have provided an important community space during COVID-19 [65] and may provide a multiplicity of wellbeing benefits [66]. Further research focusing on the typology and quality of green spaces and their relationships with mental wellbeing is warranted.

## 5. Limitations

There are several important limitations associated with this study. For example, non-random sampling methods were used, which means robust calculations of error and inferences of representativeness are not possible. It is possible that people who consider green spaces as important, and those who use green spaces, were over-represented in the sample. There was also a deficit of samples from outside of England to include in socioecological analyses and there were age and gender skews. The inclusion of additional wider-scale georeferenced samples would have provided a richer picture of socioecological dynamics. Temporally-objective information on nature exposure and analysis of seasonal influences vs. pandemic influences would also bring value. For example, as mentioned, seasonality (and the one-time sampling point) may have significantly affected our results. People are probably more likely to spend time outdoors engaging with nature during the spring and summer months (in the northern hemisphere, where the majority of samples were acquired) as the conditions are favourable for recreational activities and more flora and fauna are active during this period. We used the term “as a result of the COVID-19 pandemic” in the framing of many of our questions, and the questionnaire information sheet described how the project was a study of the behavioural responses to the COVID-19 pandemic. Future research should aim to control for this factor. The results in this study are also association based. Therefore, inferences of causation and directionality of the relationships are not possible. There are also inherent biases associated with self-reported methods and potential for responder bias, i.e., did the respondents guess what the survey was looking for and respond accordingly? Further in-depth and controlled research is warranted. A re-assessment of the data, or follow-up work could benefit from providing a deeper examination of, for example, the social structure of the sample of individuals who responded to the questionnaire and using the wellbeing instrument scores as continuous variables may provide different results (as information can be lost when recoding variables). Another limitation is that the survey was written in the English language only, and as such, only English-speaking individuals were likely to respond.

## 6. Conclusions

This study provides novel insights into the value of natural environments, particularly in response to an infectious disease pandemic. People need quality natural environments in their neighbourhoods to maintain favourable health and wellbeing. The COVID-19 pandemic has further highlighted the immense value of connecting and engaging with nature. The need for a mutually advantageous relationship between humans and the wider biotic community has never been more important. We must conserve and restore nature to maintain resilient societies and promote planetary health.

## Figures and Tables

**Figure 1 ijerph-18-02227-f001:**
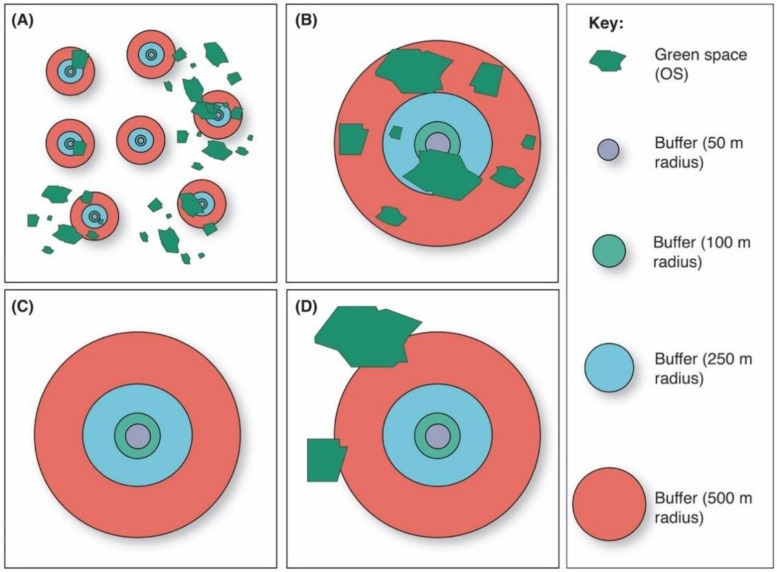
Buffer types and green space polygons used in this study. Green space shapefiles (green polygons) were imported into QGIS and buffer radii of 50, 100, 250, and 500 m were created. (**A**) shows an example where green space presence/abundance differs between buffer zones; (**B**) shows buffer zones with several green spaces within; (**C**) shows a buffer without any green spaces; and (**D**) provides an example of where green space polygons are touching the 500 m buffer but are not completely encapsulated—these would still be counted as being within this buffer zone.

**Figure 2 ijerph-18-02227-f002:**
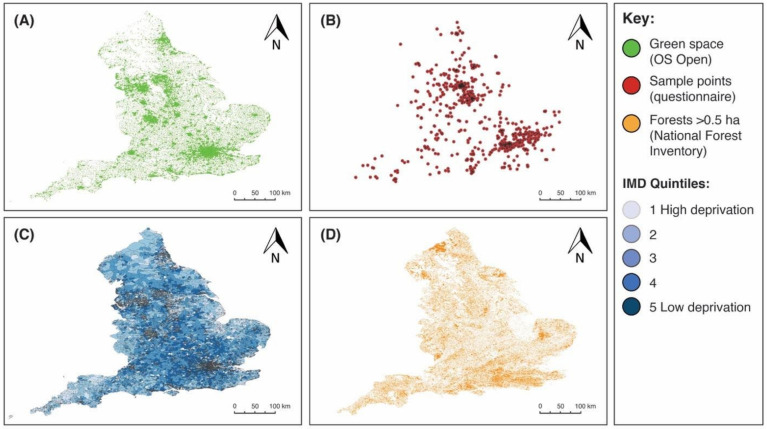
Spatial outputs combined with England boundary datasets. (**A**) shows the distribution of OS Open Green Space polygons; (**B**) shows the distribution of georeferenced samples from the survey; (**C**) shows the Lower Super Output Areas with joined index of multiple deprivation quintile data, whereby 1 corresponds to relatively high deprivation (and lighter blue) and 5 corresponds to relatively low deprivation (and darker blue); and (**D**) shows distribution of forests > 0.5 ha.

**Figure 3 ijerph-18-02227-f003:**
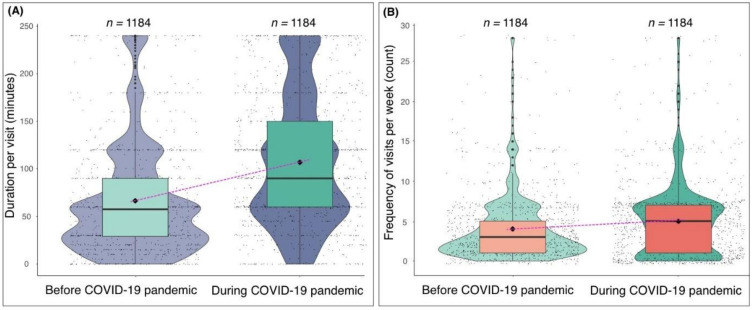
Violin plots (**A**) typical duration spent in natural environments (e.g., parks, woodland) before (**left**) and during (**right**) the COVID-19 pandemic; and (**B**) typical frequency of visits to natural environments per week before (**left**) and during (**right**) the COVID-19 pandemic. The black diamond represents the mean value.

**Figure 4 ijerph-18-02227-f004:**
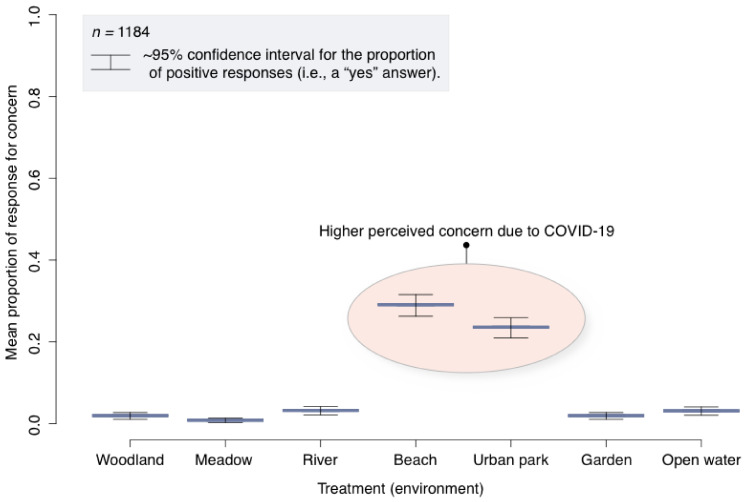
Boxplot for the GLM analysis (regarding environments of concern due to COVID-19), showing means and approximate 95% confidence intervals for the proportion of positive responses, where “yes” was recoded to “1”.

**Figure 5 ijerph-18-02227-f005:**
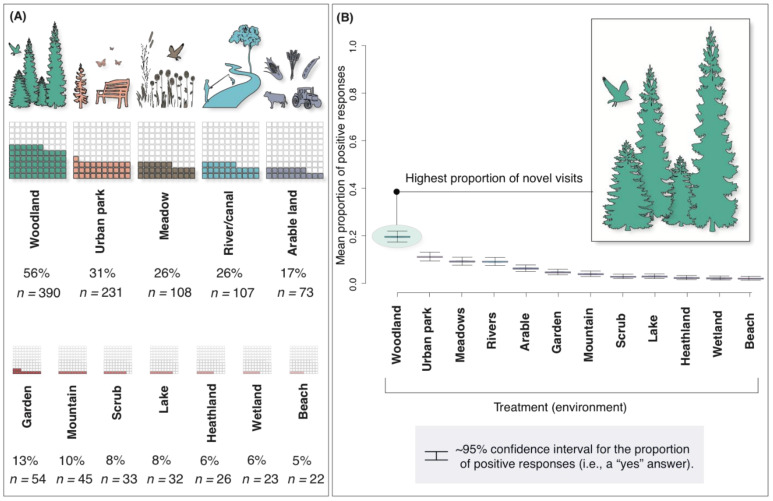
Proportions of participants who visited natural environments they would not usually visit (as a result of the COVID-19 pandemic). The top left (**A**) waffle plots show the most popular natural environments and (**B**) boxplot for the GLM analysis, shows means and approximate 95% confidence intervals for the proportion of positive responses, where “yes” was recoded to “1”.

**Figure 6 ijerph-18-02227-f006:**
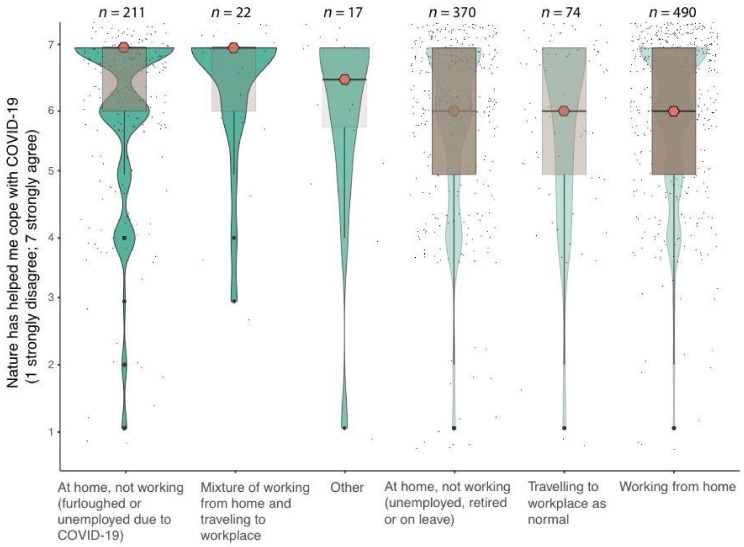
Violin plots of different Likert scores (*Y* axis) denoting level of agreement (‘nature has helped me cope with COVID-19’) analysed by home/work situations (*X* axis). Plots display median values (red diamond), interquartile range (brown) and kernel density estimation (green). The strength of the kernel colour corresponds to the median value, and the strength of the boxplot colour corresponds to the sample size.

**Table 1 ijerph-18-02227-t001:** Patterns of change in visits/exposure to natural environments as a result of the COVID-19 pandemic.

Response	No. of Responses	% of Responses
Increase in the amount of time spent in private gardens	565	47.7
Increase in the amount of time spent in woodlands	162	13.7
Increase in the amount of time spent in urban parks	129	10.9
Decrease in the amount of time spent in natural environments	71	6.0
Increase in the amount of time spent in natural environments	71	6.0
No change	60	5.1
Increase in the amount of time spent around waterbodies	49	4.1
Increase in the amount of time spent on an allotment	30	2.5
Increase in the amount of time spent at the beach	11	0.9
Decrease in the amount of time spent in urban parks	9	0.8
Increase in the amount of time spent on mountains/hills	9	0.8
Increase in the amount of time spent in meadows	8	0.7
Decrease in the amount of time spent in woodlands	4	0.3
Increase in the amount of time spent in arable land	3	0.3
Decrease in the amount of time spent around waterbodies	2	0.2
Decrease in the amount of time spent on mountains/hills	1	0.1

**Table 2 ijerph-18-02227-t002:** Differences in mean scores (before vs. during the COVID-19 pandemic) for the WEMWBS and PSS tests.

Instrument	*n*	Mean (±SD)	t	df	*p*-Value
WEMWBS before	1184	51.5 (8.2)	-	-	-
WEMWBS during	1184	44.7 (8.9)	19.1	2349	<0.01
PSS before	1184	20.9 (3.3)	-	-	-
PSS during	1184	20.6 (3.8)	1.9	2305	0.05^.^

WEMWBS: Warwick and Edinburgh Mental Wellbeing Scale (showing the mean score for mental wellbeing, where a higher score indicates higher perceived wellbeing); PSS: Perceived Stress Scale (showing the mean score for perceived stress, where a higher score indicates lower perceived stress).

**Table 3 ijerph-18-02227-t003:** Estimated regression parameters for comparisons of perceived nature-mediated coping benefits (**A**). Estimated regression parameters for comparisons of perceived nature-mediated benefits of visiting novel environments (**B**). All coefficients were significantly different from the intercept apart from the fresh air response. Perceived benefits are in descending order based on popularity of response (indicated by the mean).

**(A)** **Perceived Benefit (of Nature on Coping)**	**Mean**	**Estimate**	**Std. Error**	**z Value**	***p*-Value**
Nature provided a place to exercise	0.70	0.46	0.08	5.29	<0.01 ***
Nature helped reduce stress (Intercept)	0.60	0.40	0.05	6.84	<0.01 ***
Nature provided a calm space to think	0.58	−0.07	0.08	−0.92	0.38
Nature helped reduce anxiety	0.48	−0.48	0.08	−5.80	<0.01 ***
Nature helped provide perspective	0.46	−0.56	0.08	−6.73	<0.01 ***
Nature provided a place to be creative	0.20	−1.78	0.09	−19.04	<0.01 ***
Nature is a judgement free environment	0.18	−1.91	0.09	−19.91	<0.01 ***
**(B)** **Perceived Benefit (of Novel Environment)**	**Mean**	**Estimate**	**Std. Error**	**z Value**	***p*-Value**
Physical exercise (Intercept)	0.30	−0.82	0.06	−13.08	<0.01 ***
Fresh air	0.30	−0.05	0.08	−0.62	0.53
Relaxation	0.23	−0.37	0.09	−4.03	<0.01 ***
Reduce stress	0.20	−0.62	0.09	−6.43	<0.01 ***
Reduce anxiety	0.15	−0.91	0.10	−8.83	<0.01 ***
Space to think	0.15	−0.94	0.1	−9.08	<0.01 ***
Boost immune system	0.07	−1.77	0.13	−13.54	<0.01 ***
Beneficial microbes	0.02	−3.00	0.21	−14.20	<0.01 ***
Bathe in phytoncides (plant-based chemicals)	0.01	−3.52	0.26	−13.20	<0.01 ***

*** <0.01.

**Table 4 ijerph-18-02227-t004:** Association between NDVI, green space presence and abundance, and self-reported mental wellbeing during the COVID-19 pandemic.

	NDVI 50 m	NDVI 100 m	NDVI 250 m	NDVI 500 m	Green Space Pr 50 m	Green Space Ab 50 m	Green Space Pr 100 m	Green Space Ab 100 m	Green Space Pr 250 m	Green Space Ab 250 m	Green Space Pr 500 m	Green Space Ab 500 m
Model 1: Unadjusted †	5.14 (1.05, 25.09) ** *p* = 0.04	5.48 (1.07, 27.94) ** *p* = 0.03	8.04 (1.44, 45.01) ** *p* = 0.01	5.32 (0.95, 29.96) * *p* = 0.05	0.97 (0.58, 1.63) *p* = 0.91	1.05 (0.73, 1.49) *p* = 0.86	1.13 (0.74, 1.73) *p* = 0.58	1.0 (0.81, 1.24) *p* = 0.92	1.13 (0.61, 2.09) *p* = 0.68	0.99 (0.91, 1.08) *p* = 0.91	0.71 (0.27, 1.86) *p* = 0.50	0.96 (0.93, 1.0) *p* = 0.06
Model 2: Adjusted for gender	4.92 (1, 24.13) ** *p* = 0.04	5.26 (1.03, 26.90) ** *p* = 0.04	7.74 (1.38, 43.37) ** *p* = 0.01	5.2 (0.95, 29.3) * *p* = 0.05	0.98 (0.58, 1.67) *p* = 0.94	1.05 (0.73, 1.5) *p* = 0.80	1.16 (0.75, 1.8) *p* = 0.49	1.01 (0.82, 1.25) *p* = 0.91	1.2 (0.64, 2.24) *p* = 0.56	0.99 (0.91, 1.08) *p* = 0.96	0.83 (0.31, 2.22) *p* = 0.72	0.97 (0.93, 1.01) *p* = 0.15
Model 3: As 2 + adjusted for age	2.93 (0.56, 15.38) *p* = 0.2	3.32 (0.61, 17.93) *p* = 0.16	6.16 (1.03, 36.89) ** *p* = 0.04	4.83 (0.81, 28.87) *p* = 0.08	0.97 (0.57, 1.62) *p* = 0.89	1.04 (0.73, 1.49) *p* = 0.81	1.12 (0.73, 1.72) *p* = 0.59	1.0 (0.81, 1.24) *p* = 0.93	1.12 (0.61, 2.07) *p* = 0.70)	0.99 (0.91, 1.08) *p* = 0.91	0.69 (0.26, 1.81) *p* = 0.47	0.96 (0.93, 1.0) *p* = 0.06
Model 4: As 3 + adjusted for SES §	2.96 (0.55, 15.88) *p* = 0.2	3.39 (0.61, 18.88) *p* = 0.16	6.74 (1.07, 42.48) ** *p* = 0.04	5.42 (0.84, 35.11) *p* = 0.08	1.0 (0.59, 1.69) *p* = 0.99	1.08 (0.75, 1.54) *p* = 0.68	1.15 (0.75, 1.77) *p* = 0.52	1.02 (0.83, 1.27) *p* = 0.83	1.17 (0.63, 2.16) *p* = 0.62	1.0 (0.91, 1.09) *p* = 0.96	0.71 (0.27, 1.85) *p* = 0.49	0.97 (0.93, 1.01) *p* = 0.08
Model 5: As 4 + adjusted for nature connectedness	2.76 (0.51, 14.79) *p* = 0.23	3.15 (0.57 17.49) *p* = 0.19	6.05 (0.96, 38.11) * *p* = 0.05	4.84 (0.75, 31.35) *p* = 0.09	0.97 (0.58, 1.63) *p* = 0.91	1.04 (0.73, 1.48) *p* = 0.82	1.15 (0.75, 1.76) *p* = 0.52	1.0 (.081, 1.24) *p* = 0.93	1.17 (0.63, 2.16) *p* = 0.61	0.99 (0.91, 1.08), *p* = 0.92	0.75 (0.29, 1.97) *p* = 0.57	0.97 (0.93, 1.0) *p* = 0.06
Model 6: As 5 + living/work situation	3 (0.55, 16.46) *p* = 0.2	3.29 (0.58, 18.63) *p* = 0.17	6.08 (0.95, 38.98) * *p* = 0.05	4.56 (0.70, 29.79) *p* = 0.10	1.0 (0.59, 1.68) *p* = 0.98	1.05 (0.74, 1.49) *p* = 0.78	1.15 (0.75, 1.78) *p* = 0.51	1.01 (0.82, 1.25) *p* = 0.89	1.09 (0.58, 2.02) *p* = 0.79	0.92 (0.09, 1.08) *p* = 0.86	0.72 (0.27, 1.9) *p* = 0.52	0.97 (0.93, 1.0) *p* = 0.08
Model 7: As 6 + level of education	1.1 (096, 1.39) *p* = 0.2	3.33 (0.59, 18.74) *p* = 0.17	5.97 (0.94,37.79) * *p* = 0.05	4.71 (0.73, 30.23) *p* = 0.09	0.96 (0.57, 1.62) *p* = 0.89	1.04 (0.73, 1.49) *p* = 0.81	1.12 (0.73, 1.72) *p* = 0.60	1.0 (0.81, 1.24) *p* = 0.94	1.13 (0.61, 2.09) *p* = 0.69	0.99 (0.91, 1.08) *p* = 0.90	0.71 (0.27, 1.84) *p* = 0.49	0.97 (0.93, 1.0) *p* = 0.06
Pr = presence; Ab = abundance. Odds ratio and 95% CI reported. ** <0.05, * 0.05. † *n* = 933; § adjusted by index of multiple deprivation (IMD) quintiles; based on Nature Relatedness -6 scale (NR-6).												

**Table 5 ijerph-18-02227-t005:** Association between NDVI, green space presence and abundance, and perceived stress during the COVID-19 pandemic.

	NDVI 50 m	NDVI 100 m	NDVI 250 m	NDVI 500 m	Green Space Pr 50 m	Green Space Ab 50 m	Green Space Pr 100 m	Green Space Ab 100 m	Green Space Pr 250 m	Green Space Ab 250 m	Green Space Pr 500 m	Green Space Ab 500 m
Model 1: Unadjusted †	0.45 (0.18, 1.08) *p* = 0.07	0.38 (0.15, 0.94) ** *p* = 0.03	0.37 (0.14, 0.96) ** *p* = 0.04	0.43 (0.17) *p* = 0.08	1.06 (0.78, 1.43) *p* = 0.71	1.03 (0.84, 1.28) *p* = 0.76	0.9 (0.7, 1.15) *p* = 0.4	0.99 (0.88, 1.13) *p* = 0.98	0.87 (0.62, 1.23) *p* = 0.4	1.0 (0.95, 1.06) *p* = 0.74	0.88 (0.47, 1.65) *p* = 0.6	1.02 (1, 1.04) *p* = 0.06
Model 2: Adjusted for gender	0.5 (0.2, 1.23) *p* = 0.13	0.50 (0.17, 1.06) *p* = 0.06	0.46 (0.16, 1.06) *p* = 0.06	0.46 (0.17, 1.319 *p* = 0.10	1.08 (0.79, 1.46) *p* = 0.6	1.04 (0.84, 1.29) *p* = 0.8	0.9 (0.7, 1.16) *p* = 0.4	0.99 (0.88, 1.13) *p* = 0.9	0.85 (0.59, 1.21) *p* = 0.4	1.0 (0.95, 1.06) *p* = 0.76	0.95 (0.5, 1.79) *p* = 0.8	1.02 (1, 1.04) *p* = 0.06
Model 3: As 2 + adjusted for age	0.66 (0.26, 1.27) *p* = 0.38	0.54 (0.21, 1.38) *p* = 0.2	0.49 (0.19, 1.3) *p* = 0.15	0.52 (0.2, 1.38) *p* = 0.18	1.03 (0.76 1.41) *p* = 0.8	1.02 (0.82, 1.27) *p* = 0.8	0.86 (0.66, 1.11) *p* = 0.2	0.99 (0.87, 1.12) *p* = 0.86	0.88 (0.68, 1.16) *p* = 0.4	1.0 (0.95, 1.05) *p* = 0.86	0.84 (0.44, 1.61) *p* = 0.6	1.01 (0.99, 1.04) *p* = 0.17
Model 4: As 3 + adjusted for SES §	0.69 (0.27, 1.77) *p* = 0.43	0.55 (0.21, 1.47) *p* = 0.2	0.5 (0.18, 1.39) *p* = 0.18	0.53 (0.19, 1.5) *p* = 0.23	1.02 (0.75, 1.4) *p* = 0.87	1.01 (0.81, 1.26) *p* = 0.9	0.85 (0.66, 1.11) *p* = 0.2	0.98 (0.87, 1.12) *p* = 0.89	0.84 (0.59, 1.2) *p* = 0.4	1.0 (0.95, 1.05) *p* = 0.92	0.85 (0.44, 1.62) *p* = 0.6	1.01 (0.99, 1.04) *p* = 0.2
Model 5: As 4 + adjusted for nature connectedness	0.59 (0.23, 1.53) *p* = 0.27	0.47 (0.17, 1.25) *p* = 0.19	0.4 (0.14, 0.14) *p* = 0.08	0.43 (0.15, 1.23) *p* = 0.11	1.02 (0.74, 1.4) *p* = 0.9	0.99 (0.79, 1.24) *p* = 0.9	0.89 (0.68, 1.15) *p* = 0.3	0.99 (0.86, 1.12) *p* = 0.88	0.88 (0.61, 1.26) *p* = 0.5	1.0 (0.95, 1.05) *p* = 0.89	0.94 (0.48, 1.81) *p* = 0.8	1.02 (0.99, 1.04) *p* = 0.14
Model 6: As 5 + living/work situation	0.59 (0.23, 1.53) *p* = 0.27	0.38 (0.15, 0.94) *p* = 0.11)	0.37 (0.14, 0.96) *p* = 0.07	0.41 (0.14, 1.2) *p* = 0.10	1.02 (0.74, 1.4) *p* = 0.9	0.99 (0.79, 1.24) *p* = 0.99	0.89 (0.68, 1.16) *p* = 0.4	0.99 (0.86, 1.12) *p* = 0.89	0.89 (0.62, 1.28) *p* = 0.5	1.0 (0.95, 1.06) *p* = 0.85	0.96 (0.49, 1.85) *p* = 0.9	1.02 (0.99, 1.04) *p* = 0.12
Model 7: As 6 + level of education	0.59 (0.23, 1.53) *p* = 0.29	1.06 (0.95, 1.17), *p* = 0.3	0.39 (0.14, 1.11) *p* = 0.07	0.43 (0.17, 1.12) *p* = 0.10	1.02 (0.74, 1.4) *p* = 0.9	0.99 (0.79, 1.24) *p* = 0.98	0.88 (0.68, 1.16) *p* = 0.4	0.99 (0.86, 1.12) *p* = 0.89	0.89 (0.68, 1.16) *p* = 0.4	0.99 (0.86, 1.12) *p* = 0.89	0.96 (0.49, 1.86) *p* = 0.9	1.02 (0.99, 1.04) *p* = 0.12
Pr = presence; Ab = abundance. Odds ratio and 95% CI reported. ** <0.05 † *n* = 933; § adjusted by index of multiple deprivation (IMD) quintiles; based on Nature Relatedness-6 scale (NR-6).												

## Data Availability

All data and code used in this study are available on the *UK Data Service ReShare* at: https://reshare.ukdataservice.ac.uk/. Data Collection: 854604.

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
