# Peer review of "Nature’s Role in Supporting Health during the COVID-19 Pandemic: A Geospatial and Socioecological Study"

_ijerph, 2021, doi:10.3390/ijerph18052227_

Round 1
Reviewer 1 Report
The paper „Nature’s role in supporting health during the COVID-19 3 pandemic: A geospatial and socioecological study“ by J.M Robinson and co-workers describes shifts of people's behaviour (England) during times with restrictions to individual mobility because of the COVID-19 pandemic. This timely study applies a robust set of methodologies that is suitable in the given circumstances. The authors are aware of the limitations to their set of observations and auxiliary data. They tackle the limitations appropriately (e.g. bootstrap resampling). The text reads well; considering that statistical analysis rarely permits nice prose.
The chosen mix of observations and methods (from different disciplines) may be questioned, and it would be helpful to learn how the findings would vary depending on a given mix. However, the timeliness of the fieldwork and publication of findings justifies sticking with the given mix. Any re-assessment of the findings should consider, for example, the social structure of the sample of individuals who responded to the questionnaire. Likewise, re-analyses may be done, excluding from the analysis of the long tail of lower frequencies (table 1). Although the statistical significance (habitual tests) seems given several of the shifts of behaviour are weak, as acknowledged by the authors (e.g. line 322, 332). Therefore, using different statistical test [1] is advisable when re-analysing the observations.
Given the study's given limitations, the given conclusion “provides novel insights” (line 510) might err at the optimistic side. Nevertheless, the paper could be published in its present form expecting that the attentive and educated reader appraises the conclusion ‘cum grano salis’.
[1] Rey, William JJ. Introduction to robust and quasi-robust statistical methods. Springer Science & Business Media, 2012
Author Response
Many thanks for your feedback and for supporting the publication of this manuscript.
This is a great point. We have included a sentence at Line 623 in the limitations to acknowledge your points regarding re-analysis of different mix/deeper investigation into social structure.
We have also included other additions/changes throughout in response to the feedback received from the other five reviewers. Hopefully this also demonstrates to you that we have further considered the strengths and limitations of this study.
Reviewer 2 Report
This is a well-designed, and well-documented study that addresses important public health issues in relation to the green open spaces in the pandemic era. I recommend the article being accepted and published by the International Journal of Environmental Research and Public Health after the following questions/suggestions are addressed:
- The survey sample has consisted of an overwhelmingly proportion of female respondents, and the gender differences were identified in different sections of the results. I would be interested in reading a more in-depth discussion on nature’s health outcomes in the female population, if possible.
- The 250m radius buffer was found a threshold, which reminds me of a study examing people’s stress and their physical distances to green spaces through a Danish national survey, where 300 m and 1km are the two cutoffs. Could you try to interpret your findings and the underlying reasons, which may help evaluate the proximity between residential neighborhoods and the green open spaces, and provide evidence for future design and policy-making?
- For the Y-axis of Figure 6, did you mean “nature has helped me cope with COVID-19”?
- In Abstract Line 16-17, did you mean “There were significantly more food-growing allotments within 100 m and 250 m around respondents with high mental wellbeing scores” or “100 m and 250 m of respondents’ postcodes…?”
Thank you for your attention!
Author Response
Many thanks for your feedback. We have taken your comments on board, and have now included an additional discussion around nature’s health outcomes in the female population at Line 508 in the Discussion section. We have included another study as a contextual discussion point – one that indicates that females exhibited higher stress levels (as measured via cortisol samples) in areas with less green space, compared to male participants.
We have now included additional discussion/interpretation of our findings – particularly in relation to neighbourhood green space and the possible underlying reasons for limited wellbeing effects at proximal and distal scales. This can be found at Line 564. For example, one possible explanation for this is that people may receive additional benefits from engaging with neighbourhood green spaces (such as safe, accessible, biodiverse parks with facilities) compared to gardens or street vegetation, whereas green spaces beyond the neighbourhood range may be perceived as less accessible.
- For the Y-axis of Figure 6, did you mean “nature has helped me cope with COVID-19”?
Thanks. This has now been updated.
- In Abstract Line 16-17, did you mean “There were significantly more food-growing allotments within 100 m and 250 m around respondents with high mental wellbeing scores” or “100 m and 250 m of respondents’ postcodes…?”
Thanks. This has now been updated.
Many thanks for your constructive feedback! We have also made changes throughout following feedback from the other five reviewers.
Reviewer 3 Report
The authors address the question of whether the stress, wellbeing and outdoor activities of people in the UK changed before and after the COVID-19 pandemic using an online survey. This is an important question and very worthy of investigation. I also commend the authors on their forethought to address this question so quickly when the pandemic began. There is a lot that I like about this article: it is written very well, the statistics are rich, varied and generally appropriate and the results are clear and presented in a very appealing manner. My main concern is about the single time point data collection and the interpretation of these results. I would like to see this major concern addressed along with some other minor concerns before the paper is published.
Main concern:
Lines 104-105 state ‘We asked participants to answer questions regarding perceived stress in recent weeks, as well as in the weeks prior to the COVID-19 pandemic.’ Am I right in understanding that the authors used one time point and asked people to fill out the survey twice – once regarding their current wellbeing/stress and once regarding their wellbeing/stress prior to the pandemic? I see some problems with this approach. Can the authors detail in the Discussion how this approach may have impacted their data and conclusions and address the fact that the ‘before COVID’ data was in the middle of winter and the ‘during COVID’ data was spring/summer? At the very least I think this study and its claims should be called ‘perceived impact of COVID on wellbeing/stress’ as a more accurate reflection of the data collected using these methods. I also think that term ‘perceived’ should be used throughout in reporting any ‘changes’ in the results (currently it is used to reflect that the stress scores and wellbeing scores are ‘perceived’ but I think it should also be clear that the changes are perceived changes, not independently measured changes). Furthermore, I think some of the statements in the Discussion/Abstract should be diluted a bit such as lines 388-390 ‘Our study shows that respondents significantly changed their patterns of visiting nature as a result of the COVID-19 pandemic. People spent significantly more time…’. I think it would be more appropriate to say ‘Our study shows that respondents believed that their patterns of visiting nature significantly changed during the COVID-19 pandemic. People felt like they spent significantly more time…’. I would like to see much of the Discussion re-written to reflect this important difference, for example, 399-403, 424-437. I would also like to see a richer discussion of the way this sampling method impacted the results (the single line of 502-503 is insufficient) in the Limitations section (to me, this warrants a full paragraph in the Limitations).
Minor concerns:
Line 49 – the link with microorganisms here seems a bit out of place. Can you add a clause and reference about exposure to diverse micro-organsims benefiting mental health or wellbeing to make it fit better? E.g. ‘…regulate the human immune system and correlates with improved wellbeing’ or similar?
Lines 57-59. Can you clarify if you mean that no ‘COVID-related studies’ have incorporated socioeconomic factors, access and vegetation cover? Currently it reads like you are saying no studies have linked those factors to health and wellbeing which I would say is incorrect.
I was curious as to why the authors used aggregate measures of socioeconomic deprivation rather than individual measures since wellbeing data was collected at the individual level.
Can you provide some details of the theoretical grounding of this study and surveys? What are the theories upon which the connections between nature exposure and wellbeing or stress are based? I would like to see some of those theories made explicit which also may help guide/clarify which questions/statistical comparisons were a priori compared with post hoc.
The authors state that some of the questions were ‘pilot-tested. Can they clarify how the pilot testing informed the study design/wording of these questions to ensure they were eliciting accurate responses from participants?
There are sometimes problems around using Likert data as a continuous variable as has been done here. Can the authors state whether the assumptions of their statistical tests (e.g. normality, equal variance of residuals) has been met?
Can the authors clarify some things about the geoprocessing? Presumably the authors use the center point of the postcode and create buffers around that point? Post codes that I am familiar with are often several kilometers in area and land cover can vary substantially within a postcode. But the buffers used for the spatial analysis are a much finer-scale (50-500m). Do the authors feel the values calculated from these analyses are representative of the postcode? Can the authors comment on this? It may be appropriate to change some terminology in the article such as phrases like this on line 355: ‘spatial radii around a respondent’s home location’
I like Figure 1 – very clear and helpful in displaying the green space methods.
How was the data rectified across scales (e.g. deprivation data with the postcode data)?
Lines 188-189 say ‘We applied model adjustments for gender, age, socioeconomic 188 status, level of education, work/living situation, and nature connectedness.’ Can you provide more details of how you made these adjustments?
Lines 213-217 state that this study found that respondents spent more time in greenspaces ‘as a result of COVID’. Can the authors contextualise this statement a) without implying causation, b) regarding lockdown measures in the UK, and c) regarding socioeconomic regions of the country?
I am surprised that IMD did not affect the amount of time spent in nature, given the general pattern that people with lower socioeconomic areas have less access to green spaces. Can the authors explain and present the range of IMDs captured in the survey (from the map it seems like it might be skewed towards higher socioeconomic areas but it is hard to tell) and the impact this may have had on the outcome of the study?
Lines 257-258 (and 406-407) say ‘This implies that concern for contracting SARS-CoV-2 virus may influence people’s decision to spend time in certain environments.’ I don’t think you can make that claim given the data. Especially considering Figure 5 shows that urban parks were second highest in novel visits. Can you justify this claim or reword?
The authors have report on a significant number of tests, given the number of survey responses in a way that might increase the likelihood of false positives. Can the authors clarify which questions were planned a-priori and which were conducted post-hoc and whether the post-hoc tests received any additional statistical correction?
I think care is needed in interpreting the ‘reduced stress of furloughed people’ (e.g. lines 285-286, 430-437). Did this reduction occur in individuals with lower education levels (typically associated with lower socioeconomic levels) or people living in areas of higher disadvantage (if these were sufficiently sampled). I think it is important not to over-generalize this finding especially to socioeconomic levels which were not represented and to other circumstances (e.g. countries where government support wasn’t assured or for migrants in a country that didn’t receive government support).
Lines 461-462 – I’d be careful with this interpretation – isn’t it equally likely that people with high nature connectedness score are more likely to spend time in nature? Please rephrase.
I am not convinced by the discussion on lines 469-480. Why would greenness significance at 250m lead to the conclusion that ‘very proximate, neighbourhood scale, land-cover greenness ‘ is important when more proximate (50, 100, 150) and neighbourhood-y (500m) greenness was not significant? I think a more cautious interpretation that includes the possibility of spurious relationships from over-sampling a dataset would be appropriate.
Table 2 – how is ‘public park or garden’ different from ‘public park’?

Author Response
Many thanks for your constructive feedback. We have taken your comments on board and have made several changes to the manuscript to reflect this.
For example, we have now included the term ‘reportedly’ or ‘perceived’ throughout to show that the results are self-reported and not necessarily spatially and temporally explicit using objective measures.
We have also included an additional discussion regarding the seasonality and one-time point issue at Line 613 in the Limitations section.
We have also included additions throughout the manuscript in response to the other five reviewers.
Reviewer 4 Report
Review of the manuscript ijerph-1084783
The manuscript titled as “Nature’s role in supporting health during the COVID-19 pandemic: A geospatial and socioecological study” reports a mixed-method study including the analysis of web-based questionnaires and spatial analyses using geographic information system (GIS). Overall, the topic is very timely and important. The study has been carefully conducted using validated measures and statistical analyses, and reporting is clear. I have some minor comments that could improve the manuscript:
Abstract
- The size of the data could be reported in Abstract. Please consider also mentioning that most of the respondents were from England, or that the study was conducted in England (many of the results are only from England).
Introduction
- 2, line 58: Deprivation of what, please specify. This comment applies throughout the manuscript where deprivation is discussed.
- 2, lines 60-61: Please specify that you analyzed changes before and during the COVID-19 pandemic.
- 2, lines 63-70, section starting with “We use online pilot-tested questionnaires…”: This section is about methods and therefore it is not needed in Introduction. Please consider moving it to Methods section and if they are already mentioned there, it can also be removed.
Methods
- 3, line 121: “The questionnaire was distributed across the world…” “World” "World" sounds like that you did a global study. However, almost all responses are from England. Were the methods for reaching participants located in England, and if they were outside England, where they were? Please specify how they were geographically located. In addition, the questionnaire was only in English so only English-speaking individuals could respond.
- 3, line 126: It seems that also geographical location was used as an exclusion criterion because the results are reported only from England even if you got responses also from outside England.
- 5, line 173: A repeated-measures ANOVA would have been a more suitable method because you analyzed changes between two time points.
- 5, line 179: Why were WEMWBS and PSS scores not used as continuous variables? Please provide an explanation. Information is always lost when a variable is recoded into fewer categories.
Results
- 5, line 199: Did you exclude responses from outside England, UK?
- 5, line 204: Was the sample representative of the UK population in terms of age?
- Tables with p-values (eg. Table 2, 3, 4, 5): Why are both exact p-values and stars reported? Only one of them is sufficient and saves space. This comment applies to all tables including p-values.
Discussion
- 14, lines 419-421: “Our results indicate that woodlands were the most popular novel environment with 56% of these respondents visiting woodlands when they would not usually.” This sentence ends in a strange way, please check.
Limitations
- Another limitation is that due to non-random sampling it is likely that people who consider green spaces as important, and those who use green spaces, were over-represented in the sample. Please consider its role in the interpretation of the results in the Limitations section.
Author Response
The size of the data could be reported in Abstract. Please consider also mentioning that most of the respondents were from England, or that the study was conducted in England (many of the results are only from England).
We have included this at Line 16 and 18 in the Abstract.
line 58: Deprivation of what, please specify. This comment applies throughout the manuscript where deprivation is discussed.
Many thanks for this. We have now included a definition of deprivation at Line 61, and Line 201. For example, in this study the Index of Multiple Deprivation measure takes into account some of the following to define relative deprivation: economic factors, crime risk, education and living environment.
lines 60-61: Please specify that you analyzed changes before and during the COVID-19 pandemic.
Thanks. We have now specified this at Line 68 in the Introduction.
lines 63-70, section starting with “We use online pilot-tested questionnaires…”: This section is about methods and therefore it is not needed in Introduction. Please consider moving it to Methods section and if they are already mentioned there, it can also be removed.
We have kept the brief methodological overview in the Introduction. We feel it is only brief and provides an indication of methods for context, which are described in greater detail in the methods section.
line 121: “The questionnaire was distributed across the world…” “World” "World" sounds like that you did a global study. However, almost all responses are from England. Were the methods for reaching participants located in England, and if they were outside England, where they were? Please specify how they were geographically located. In addition, the questionnaire was only in English so only English-speaking individuals could respond.
Thanks for this feedback. We have included clarification at Line 139. We have included the English -speaking limitation in the Limitations section at Line 627.
line 126: It seems that also geographical location was used as an exclusion criterion because the results are reported only from England even if you got responses also from outside England.
Thanks for this point. This has now been clarified.
line 173: A repeated-measures ANOVA would have been a more suitable method because you analyzed changes between two time points.
Thanks for this suggestion. We will take this on board for future studies.
line 179: Why were WEMWBS and PSS scores not used as continuous variables? Please provide an explanation. Information is always lost when a variable is recoded into fewer categories.
We used cut-off scores as one suggested approach from the instrument designs and/or other protocols. We have acknowledged that the continuous variable approach may reveal different findings in the Limitations.
line 199: Did you exclude responses from outside England, UK?
Thanks. We have now clarified this at Line 247.
line 204: Was the sample representative of the UK population in terms of age?
We have now included a line here to suggest that the age was not quite representative of the current UK age structure with reference to data acquired from the Office of National Statistics (Line 252).
Tables with p-values (eg. Table 2, 3, 4, 5): Why are both exact p-values and stars reported? Only one of them is sufficient and saves space. This comment applies to all tables including p-values.
Due to the lack of asterisks (because there were only limited statistically significant results) we have decided to keep the asterisks to emphasise where there are statistically significant results.
lines 419-421: “Our results indicate that woodlands were the most popular novel environment with 56% of these respondents visiting woodlands when they would not usually.” This sentence ends in a strange way, please check.
We have now refined this sentence.
Another limitation is that due to non-random sampling it is likely that people who consider green spaces as important, and those who use green spaces, were over-represented in the sample. Please consider its role in the interpretation of the results in the Limitations section.
This is a good point. We have now included this in the limitations along with several other comments in response to the feedback from the other five reviewers.
Reviewer 5 Report
The paper presents a very interesting read. The study is very interesting and well conducted. I suggest to publish the after some minor revisions.
Line 86-88: The statement sounds a bit odd: I´d suggest to shift from a statement with “we belief“ to something based on evidence e.g. citing one or two references of literature of the demonstrated benefits of accessible nature or greenspace and it´s multiple benefits for health or other values around it, not only related with a narrow focus on pandemics. I´d suggest e.g. using a kind of broader review or overview article on nature or greenspace and health benefits as a reference for this statement.
Line 120-124: Methodology: The description for this is a bit short and it would be helpful for a sound methodological understanding explaining your starting points and strategies for spreading your questionnaires a bit better to follow up to ensure replicability. I consider it very important to add some information about the strategies and networks used. A short information in the main text and an appendix with a list of networks and persons (anonymised if and where needed) used to spread would be fine.
Discussion section: An aspect missing would be a bit more discussion in a short section where you compare and involve also reflection on benefits of nearby nature and/or greenspace also in non-pandemic times to compare your findings also in a broader context with some literature.
Author Response
The paper presents a very interesting read. The study is very interesting and well conducted. I suggest to publish the after some minor revisions.
Many thanks for your feedback.
Line 86-88: The statement sounds a bit odd: I´d suggest to shift from a statement with “we belief“ to something based on evidence e.g. citing one or two references of literature of the demonstrated benefits of accessible nature or greenspace and it´s multiple benefits for health or other values around it, not only related with a narrow focus on pandemics. I´d suggest e.g. using a kind of broader review or overview article on nature or greenspace and health benefits as a reference for this statement.
This is a good point. We have now revised the line to suggest that conserving nature is important for overall health and have cited a systematic review on the health benefits of engaging with nature/green space.
Line 120-124: Methodology: The description for this is a bit short and it would be helpful for a sound methodological understanding explaining your starting points and strategies for spreading your questionnaires a bit better to follow up to ensure replicability. I consider it very important to add some information about the strategies and networks used. A short information in the main text and an appendix with a list of networks and persons (anonymised if and where needed) used to spread would be fine.
Thanks for this feedback. We have now included additional information on this at Line 140. Furthermore, as suggested, we have included our strategy in the Supplementary Materials at Line 666.
Discussion section: An aspect missing would be a bit more discussion in a short section where you compare and involve also reflection on benefits of nearby nature and/or greenspace also in non-pandemic times to compare your findings also in a broader context with some literature.
Many thanks for this suggestion. We have included a short additional passage to reflect broader benefits and also a potentially interesting line of enquiry by comparing COVID-19 era observations with those in post-pandemic times.
Reviewer 6 Report
Comments to authors
This is a great opportunity to review your paper, “Nature’s roles in supporting health during the COVID-19 pandemic: A geospatial and socioecological study.” The authors examined the role of nature in supporting health during the COVID-19 pandemic by using a web-based questionnaire and spatial analysis. It is a very interesting paper. However, I have some comments and questions.
- The introduction is good but it is not sufficient, particularly about roles of nature during the pandemic. The authors should add some literature focusing on the roles and patterns of nature in public health outcomes during the pandemic.
- Line 120-121, p3: the authors indicated that the web-based questionnaire was distributed across the world, but Line 197-199, p5 the authors found that 96% of the georeferenced responses were predominantly from across England. In line 346-348, the authors showed that 94%the survey responses came from the UK. Doubting how the authors distributed the weblink? Why the web link could not reach other populations of the world? Since it was a UK dominant response, could the result reflect only the UK’s situation/condition rather than other parts of the world? Furthermore, it would be interesting to see where 4% of the responses were from.
- Table 4 and 5 were hard to read because it was so messy.
- Line 389-390: the authors indicated that “People spent significantly more time in nature and visited nature more often during the pandemic.” Line 399-341, “As a result of the COVID-19 pandemic, over 90% of respondents increased the amount of time they spent in natural environment….” These statements may be applicable for other contexts, particularly in some countries where the governments have restricted/banned a visit to public parks or greenspaces during the pandemic. So, it is better to confirm that the responses of this study reflect only the UK’s context although the authors attempted to collect data from countries around the globe.
- Line 418-423: What were the reasons for a shift from visiting environments they used to visit new ones? Was it because of the COVID-19 pandemic? Would it be a reason that the respondents may have more time to take an adventure as they may have been laid off or their business has been suspended as a result of the pandemic?
- The roles of nature in public health-related outcomes were well-studied. Of course, a few studies explore the association between nature and public health during the pandemic. In general, the recommendations to conserve and restore nature to maintain resilient/green communities/cities do not work for cities/communities where space is limited (or not available) as a result of uncontrolled sprawl. Therefore, it would be more interesting to read suggestions from your study on how to restore nature for cities where green spaces/parks are underprovided.
- Table 1 in the supplementary files was not good to read. Please check and redesign it.
Author Response
This is a great opportunity to review your paper, “Nature’s roles in supporting health during the COVID-19 pandemic: A geospatial and socioecological study.” The authors examined the role of nature in supporting health during the COVID-19 pandemic by using a web-based questionnaire and spatial analysis. It is a very interesting paper. However, I have some comments and questions.
- The introduction is good but it is not sufficient, particularly about roles of nature during the pandemic. The authors should add some literature focusing on the roles and patterns of nature in public health outcomes during the pandemic.
Thanks for your feedback. We have taken your comments on board and have included another example of a recent study exploring the roles of nature during the COVID-19 pandemic. However, we feel there is now a sufficient amount of context in the manuscript.
- Line 120-121, p3: the authors indicated that the web-based questionnaire was distributed across the world, but Line 197-199, p5 the authors found that 96% of the georeferenced responses were predominantly from across England. In line 346-348, the authors showed that 94%the survey responses came from the UK. Doubting how the authors distributed the weblink? Why the web link could not reach other populations of the world? Since it was a UK dominant response, could the result reflect only the UK’s situation/condition rather than other parts of the world? Furthermore, it would be interesting to see where 4% of the responses were from.
Thanks for this feedback. We have provided more clarification regarding the distribution of the questionnaire now throughout the manuscript, for example, at Line 140.
Great point regarding the participation strategy. We have also included a protocol in the Supplementary Material to further describe our approach (which we acknowledge has its limitations) at Line 666.
- Table 4 and 5 were hard to read because it was so messy.
Thanks for this. We have adjusted the tables so that there are vertical and horizontal borders inside – this hopefully helps to separate the data.
- Line 389-390: the authors indicated that “People spent significantly more time in nature and visited nature more often during the pandemic.” Line 399-341, “As a result of the COVID-19 pandemic, over 90% of respondents increased the amount of time they spent in natural environment….” These statements may be applicable for other contexts, particularly in some countries where the governments have restricted/banned a visit to public parks or greenspaces during the pandemic. So, it is better to confirm that the responses of this study reflect only the UK’s context although the authors attempted to collect data from countries around the globe.
We feel that we have now emphasised the UK-centric aspect of the study throughout the manuscript, including in the Methods, Results, Discussion and Limitations.
- Line 418-423: What were the reasons for a shift from visiting environments they used to visit new ones? Was it because of the COVID-19 pandemic? Would it be a reason that the respondents may have more time to take an adventure as they may have been laid off or their business has been suspended as a result of the pandemic?
This is a good point and yes, this could be a reason – but having more time, being laid off or having their business suspended can, in most cases, be attributed to the disruption/s caused by the COVID-19 pandemic. We feel that we have provided sufficient context and discussed the limitations of our study in the manuscript. Furthermore, many of our questions were framed with a ‘as a result of the covid-19 pandemic, did you change X,Y,Z?’ approach.
- The roles of nature in public health-related outcomes were well-studied. Of course, a few studies explore the association between nature and public health during the pandemic. In general, the recommendations to conserve and restore nature to maintain resilient/green communities/cities do not work for cities/communities where space is limited (or not available) as a result of uncontrolled sprawl. Therefore, it would be more interesting to read suggestions from your study on how to restore nature for cities where green spaces/parks are underprovided.
Thanks for this feedback. We have now included a short discussion on the need to develop innovative strategies to optimise the design, conservation and restoration of green spaces in urban areas where space is limited due to urban densification –– and we’ve included a few initial suggestions. This can be found at Line 570.
- Table 1 in the supplementary files was not good to read. Please check and redesign it.
Thanks for pointing this out. We appreciated that the table contains an abundance of information. It is difficult to present this in a pleasing manner – particularly as the journal does not appear to allow a landscape orientation to spread the data out. As with Table 4 and 5, we have now added vertical and horizontal borders inside the table in the Supplementary Material – this hopefully helps to separate the data.
Round 2
Reviewer 6 Report
Dear Authors
The authors addressed all my comments and thus I satisfied the present form of the manuscript and recommended for acceptance and publication.